# From Green Spaces to Squares: Mapping the Distribution of Taijiquan Organizations in London

**DOI:** 10.3390/ijerph18168452

**Published:** 2021-08-10

**Authors:** Peng Zhang, Yan Jin, Zhan Gao, Xiujie Ma

**Affiliations:** 1School of Wushu, Chengdu Sport University, Chengdu 610041, China; zhangp5106@163.com; 2School of Foreign Languages, Chengdu Sport University, Chengdu 610041, China; goldenjy@126.com; 3School of Journalism, Media and Culture, Cardiff University, Cardiff CF10 1FS, UK; GaoZ3@cardiff.ac.uk; 4Chinese Guoshu Academy, Chengdu Sport University, Chengdu 610041, China

**Keywords:** London, Taijiquan, spatial distribution, cultural communication, influencing factors, accessibility

## Abstract

Contributing to Taijiquan studies, this research uses spatial analysis tools in ArcGIS 10.3 and SPSS 23.0 to map out the spatial distributional pattern of the Taijiquan organizations in London, and then explores factors attributing to the spatial distribution of Taijiquan culture. The result shows that the distribution of Taijiquan organizations in London generally presents a spatial distribution structure of “dense center + sparse periphery”; the spatial distribution is unbalanced, showing a cohesive distribution; the directional distribution tends to be obvious in areas that are proximate to urban traffic arteries and afforestation in London. Through multivariate hierarchical regression analysis, the study explores the influential factors for the spatial distribution of Taijiquan organizations in London. The results show that: population size, economic level, and education level have little influence on the spatial distribution of Taijiquan organizations; however, the population density of people over 65 years old, the accessibility of public service facilities such as green spaces, and public urban traffic has a significant impact on the spatial distribution of Taijiquan organizations.

## 1. Introduction

Known as the “Chinese Wushu Treasure” in its homeland of China, Taijiquan has attracted international academic attention for its profound cultural heritage, soft and unique fitness forms, and health benefits. Studies have investigated Taijiquan from many perspectives, especially from the perspectives of biomedicine and human health, and mainly regarded Taijiquan as a supplement and alternative medicine [1]. In spite of the papers on health-related aspects of Taijiquan, there is still no Taijiquan program as a complementary and alternative medicine widely and systematically applied in real social life. For example, Harmer argues that only if more researchers get involved in Taijiquan and promise to study Taijiquan’s health benefits systematically that, in the long run, the research results about Taijiquan will become widely applied and promoted [2]. Besides biomedical studies and health studies about Taijiquan, the popularization and globalization of Taijiquan are required. Additionally, as Ryan mentions, the future identity of Taijiquan will depend on the ways the technical and cultural control is negotiated between continents and on the interest shown by the global scientific community in the value of Taijiquan for understanding health and well-being. [3]

In December 2020, UNESCO inscribed Taijiquan on the Representative List of the Intangible Cultural Heritage of Humanity. In a certain way, the inclusion of Taijiquan in the UNESCO list has boosted the globalization of Taijiquan and the promotion of Chinese culture and sport. For instance, UNESCO has implemented a five-year protection plan for Taijiquan (2021–2025) and set up the Taijiquan Protection Coordination Committee [4]. As Taijiquan is experiencing faster globalization (transformation) than before, reasonable global studies on Taijiquan are of significance to understand Taijiquan “globalizing the local” and “localizing the global”, which is a response to the need of constructing a globalization theory for China’s indigenous sports culture.

According to the literature review, there are differences between academic research projects on Taijiquan in and out of China. On one hand, it is found that most Chinese studies have worked on the theoretical construction, strategies, and countermeasures, and preceded research for Taijiquan international communication, and interpreted Taijiquan as an international product of culture and diplomacy in the context of Chinese state policies [5,6,7,8,9,10,11,12,13], rather than in the global context. On the other hand, researchers out of China have highlighted the evaluation of Taijiquan’s health benefits in biomedical and health studies [14,15,16,17,18,19,20], with little interest in the role of Taijiquan in the humanities and social sectors. Up to now, only a few scholars have made tentative research on the globalization, transformation, and adaptation of Taijiquan [3,21]. It is evident that academic concerns about Taijiquan studies outside China are fueled by globalization.

In aiming to fulfill the global study on Taijiquan, we differentiate globalization from internationalization for Taijiquan. They are two concepts of homogeneity. The former refers to transnational processes involving interconnections among people around the world beyond national boundaries in automatic, localized, and fused styles; meanwhile, the latter focuses on enhancing relationships between countries and regions through channels, organizations, and target transmission of Taijiquan [22]. The global study on Taijiquan addresses the spatial formation and the localization of Taijiquan outside China with some approaches from human geography and analyzes the spatial distribution of Taijiquan organizations in London and the main influencing factors for the distribution (Taijiquan organization in the study is used to refer to all Taijiquan communication agencies, Taijiquan clubs, Taijiquan training institutions, schools and so on). For instance, Zhang Jie studied the spread and cultural influence of Taijiquan in the United States and found that there were some regional differences in the number of Taijiquan organizations there, which were affected by such factors as population number, population density, income, and age. He also concluded that the population number and percentage of elderly and middle-aged persons are positively related to the number of Taijiquan organizations in the regions he investigated [23]. Various studies reporting associations between urban green space and health and well-being proposed that good quality open space is related to a better quality of life for urban residents. [24] Public green space can improve the sporting attainability for people living near the area [25,26,27,28,29]. Besides, there are associations between green space, health, and well-being [30,31], and the accessibility, area, and quantity of park green space are positively correlated with the residents’ overall health perception [32,33,34]. Green common space also plays a role in the formation and maintenance of social relations among the elderly [35], and the improvement of the survival rate of the elderly [36]. Therefore, this study also aims to explore the relationship between public green space and Taijiquan organizations.

The study uses spatial analysis methods to explore the spatial distribution law and indicators of Taijiquan transmission, such as location, block, distance, accessibility, and spatial relationship [37] to provide reasonable spatial layout guidance for Taijiquan transmission, as well as the globalization of other excellent Chinese traditional cultural pursuits.

## 2. Research Area and Data Source

### 2.1. Research Area Selection

Since the 18th century, London has been one of the world’s most important political, economic, cultural, art, entertainment, and sports event centers [38], which has had a highly-developed, market-oriented sports economy and inclusive and diversified culture with a variety of sports events and activities. London has been one of the top three in the World’s Most Comprehensive City Ranking for nearly ten years in a row, ranking No.1 in Kearney 2020 Global Cities Outlook results [39]. In the Kearney 2019 Global Cities Report, London is listed as a top leader in the Global Cities Index (GCI) metrics by dimensions of cultural experience and sports events [40].

Taijiquan, which is different from local and indigenous sports in Great Britain, is said to have been taught in London since the 1940s. In 1948, Chee Soo founded the Manor Road School in West Ham, London, and taught “Li style Taijiquan” along with Daoist ideas [41]. In 1960, Gerda Geddes (considered to be the first European to bring Taijiquan to the United Kingdom) officially taught a recognized style of Taijiquan with demonstrations at the London Contemporary Dance School and disseminated the art among a vast number of British people through a TV program [42]. In 1970, John Kells, a Yang Shouzhong’s disciple, set up the British Tai Chi Chuan Association in Upper Wimpole Street, London, and offered Taijiquan courses for over 10,000 students between 1977 and 1993 [43]. In 1983, Gary Wragg founded the London Centre of Wu Style Tai Chi Chuan and taught Wu style Taijiquan [44]. In 1984, Dan Docherty, Chairman of the Tai Chi Union for Great Britain (TCUGB), founded the Practical Tai Chi Chuan School in London after his return to the UK from Hong Kong and started to teach and promote Wudang Taijiquan routines, equipment, Sanshou training, and internal exercises [45]. In 1986, Li Shaoqiang (Rose Li) began teaching Taijiquan in London from Durham and did not stop until her death in 2001 [45].

Since the founding of the Tai Chi and Qigong Union for Great Britain in 1991, British Taijiquan organizations have steadily developed under a national recognition framework with self-management models [45]. Thus, we selected London as the study area. According to the Office for National Statistics [16], London, with an area of 1572 km^2^, is divided into 33 boroughs in Inner London and Outer London (Figure 1).

### 2.2. Data Source and Processing

We use data and resources from the official website of the Tai Chi and Qigong Union for Great Britain (TCUGB) [46] to identify the number and coordinates of Taijiquan organizations in London and used Google Earth to create vector maps for their spatial distribution. We collected information about traffic route lines and public green space collected from the Ordnance Survey [47] and information about population, economic and educational statistics in various boroughs of London from the British government networks [48], including the Office for National Statistics [49], and made the vector maps utilizing ArcGIS.

The governing body (TCUGB) provides links or contact information to all the registered Taijiquan organizations. We selected the area of London on the website and collected the addresses and zip codes of Taijiquan organizations in London. With the aid of Google Earth, we obtained 208 coordinates of Taijiquan organizations in London (Table 1). The collection was done in December 2020. Consequently, we converted the valid zip codes of Taijiquan organizations into geographic coordinates and obtained the geographic spatial position of Taijiquan organizations in London, and, finally, established the database for the spatial distribution of Taijiquan organizations in London.

## 3. Research Methods

### 3.1. GIS Analysis for the Spatial Distribution of Taijiquan Organizations in London

With the aid of ArcGIS10.3 software, the study aims to (1) make a comprehensive consideration about the distribution pattern, agglomeration, and distribution density of Taijiquan organizations in terms of the nearest neighbor index, geographic concentration index, and kernel density estimates; (2) apply buffer analysis [50] to the relationship between the spatial distribution of Taijiquan organizations and traffic from main roads and public green space; (3) explore the center and development of the spatial distribution of Taijiquan organizations with standard deviation ellipse through Directional Distribution in the software [50].

#### 3.1.1. Nearest Neighbor Distance and Nearest Neighbor Index

Nearest neighbor distance (NND) is the distance between the centroid position of each element and its nearest neighboring element. The nearest neighbor index (NNI) is expressed as the ratio of the observed mean distance to the expected mean distance. The expected NND is the average distance between neighbors in a hypothetical random distribution [51]. NNI is an indicator that measures two punctiform elements adjacent in their geological space. It is normally used to analyze the point distributions, which have three types—random, uniform, and agglomerated distribution [52].

When the point distribution in the area is random (Poisson distribution), the nearest distance ${\bar{r}}_{E}$ can be expressed as follows:(1)$${\bar{r}}_{E}=\frac{1}{2\sqrt{D}}=\frac{1}{2\sqrt{m/A}}$$
where ${\bar{r}}_{E}$ is the average expected distance for a random distribution, *m* is the number of points in the map, A is the area, D is a point density. R is the nearest neighbor index. The nearest neighbor index R can be expressed as
(2)$$R=\frac{{\bar{r}}_{1}}{{\bar{r}}_{E}}=2\sqrt{r_{1}D}$$

In the formula, ${\bar{r}}_{1}$ is the average distance for a random distribution. At the time R>1, when the spatial distribution of Taijiquan organizations is uniform and balanced. When R<1, it indicates that the spatial distribution of Taijiquan organizations is aggregated. When R=1, the spatial distribution of Taijiquan organizations is randomized [53].

#### 3.1.2. Geographical Concentration Index

Geographical concentration index (GCI) is an important indicator that reflects the geographical concentration of research objects [54]. This paper uses its metrics to organize spatial distribution in London. The formula is:(3)$$G=100\sqrt{\sum_{i=1}^{n}{\left(\frac{X_{i}}{T}\right)}^{2}}$$
where G is the geographic index of Taijiquan, $X_{i}$ is the number of Taijiquan organizations in the i-th borough of London, T is the total number of Taijiquan organizations, n is the total number of boroughs in London. When G ranges from 0 to 100, the greater the value of G, the more centralized the distribution of Taijiquan organizations in London is. The smaller the value of G, the less centralized the distribution of Taijiquan organizations in London is [55].

#### 3.1.3. Kernel Density Estimation

Kernel density estimation (KDE) is a method for checking the distribution characteristics of the research elements in the overall spatial density. It can clearly reflect the aggregation of research elements from the layer. The formula is:(4)$$f\left(x\right)=\frac{1}{Th}\sum_{i=1}^{T}k\left(\frac{x-X_{i}}{h}\right)$$
where f(x) is the kernel density estimation, $k\left(\frac{x-X_{i}}{h}\right)$ is the kernel function, T is the number of Taijiquan organizations, h>0 is the bandwidth, $\left(x-X_{i}\right)$ is the distance from the estimation point x to the actual point $X_{i}$. The greater the kernel density estimate is, the more intensive the density of the Taijiquan organizations is, the higher the possibility of Taijiquan events is [56].

#### 3.1.4. Buffer Analysis

Buffer Analysis is a method to calculate the number and area of urban public service facilities within a certain radius or to calculate the number and area of buildings (such as residential blocks) within a certain radius of urban public service facilities [57]. The study applies buffer analysis to calculate the number of Taijiquan organizations within the buffer zones near public traffic arteries and public green space in London.

#### 3.1.5. Direction Distribution Analysis

Direction distribution is used to calculate the standard distance between two focal points in the directions of X and Y with the average center of the set of points as the starting point. The axis of the elliptical value is defined with two measurements. The ellipse is referred to as the standard deviation ellipse. The center of the ellipse is the spatial distribution center and the long axis of the ellipse directs the spatial development of the point data [58,59].

### 3.2. Multivariate Hierarchical Regression Analysis for the Spatial Distribution of Taijiquan Organizations in London

Multivariate hierarchical regression analysis is a statistical technique that uses several explanatory variables to predict the outcome of a response variable. It enables us to study the individual influence of these variables on yield and model the linear relationship between the explanatory variables and response variables. The demographic variables in the study include population density, the population density of people over 65 years old, average education level, and per capita GDP level, which are not subject to any explanatory variables, while the human geographic variables such as urban traffic accessibility and public green space accessibility are highly correlated and may be subject to explanatory variables. Therefore, these six variables can be divided into two blocks. The regression coefficient can be calculated by applying the block of demographic variables in the multiple regression model and the predictions about urban traffic accessibility and public green space accessibility can be calculated by applying the block of human geographic variables in the multiple regression model. The interdependence between demographic variables and human geography can affect the number of Taijiquan organizations. Finally, the prediction about the interactive variables can be calculated by applying the Average education level urban traffic accessibility and Average education level. Public green space accessibility and Average education level * Urban traffic accessibility * Public green space in the multivariate regression model.

## 4. Spatial Distribution Characteristics of Taijiquan Organizations in London

### 4.1. The Overall Spatial Distribution Structure of Taijiquan Organizations in London

The overall spatial distribution density of 208 Taijiquan organizations in London is analyzed through the tool, Kernel Density Estimation in ArcGIS10.3. It presents a high-density circle with inner London as the kernel and a low-density circle in outer London. In other words, it presents a spatial distribution structure of a “dense center + sparse periphery” (Figure 2). The uneven spatial distribution indicates that Camden and Islington are the kernel area of the high-density circle for Taijiquan organizations and there are fewer Taijiquan organizations in the peripheral area. The ratio of 208 Taijiquan organizations in inner London and outer London is 127: 81, also reflecting the density of Taijiquan organizations in the central part of London, where the number of Taijiquan organizations in Camden and Islington respectively account for 12.02% and 7.21% of the total number in London.

The directional distribution analysis shows the kernel density of Taijiquan organizations in London is located in Hyde Park in the City of Westminster (Figure 3). The standard deviation elliptical area shows the density area of Taijiquan organizations in London, where there are 145 organizations, accounting for 69.71% of the total. The rotation angle of the standard deviation ellipse is 112° with the elliptical long axis orienting from the northwest-southeast, which indicates that there are more Taijiquan organizations in the northern-southern areas of London and fewer in the eastern-western areas of London. The small flattening of the standard deviation ellipse indicates that the spatial distribution of Taijiquan organizations in London tends to expand in the direction of east and west.

### 4.2. Unbalanced Spatial Distribution of Taijiquan Organizations in London

The average distance of expected spatial distribution for the 208 Taijiquan organizations in London is 1524 m, and the actual average distance calculated with ArcGIS10.3 software is 2247 m with its nearest neighboring point index R-value reaching 0.68, which shows that the spatial distribution of Taijiquan organizations in London is an agglomerated type, showing a cohesive distribution. The geographic concentration index G of 208 Taijiquan organizations in London is 22.05, which is much higher than the average index G value of even distribution in all boroughs (G value 6.3), which indicates that the distribution agglomeration of Taijiquan organizations in London is high in Camden, Islington, Barnett, and Wandsworth (Figure 4).

### 4.3. Unclear Directivity of Taijiquan Organizations along the Traffic Arteries in London

The study sets up a buffer zone of 200 m on both sides of the urban main traffic arteries for buffer analysis with ArcGIS10.3. It is found that there are 117 Taijiquan organizations located near or along the urban main traffic arteries in London (Figure 5), accounting for 56.25% of the total. It indicates that Taijiquan organizations in London are not clearly distributed for access to the urban main traffic arteries.

### 4.4. Clear Directivity to the Public Green Space for the Distribution of Taijiquan Organizations in London

It is considered that the threshold value of people’s walking time is 15 min, and the corresponding walking distance threshold is 900 m [60]. As the actual walking distance is different from the linear distance of the buffer zone, the study uses the linear distance to determine the buffer range of public green space (This study uses public green space to refer to such areas as parks, green spaces, squares, sports public venues, etc.). The non-linear coefficient (the ratio of the shortest walking distance between the two points and the linear distance) generally ranges from 1.2 to 1.4, indicating the average walking distance being about 700 m [61].

The study uses the buffer analysis tool of the ArcGIS 10.3 with a [0-9]{1,}-m distance as a buffer radius to identify the number of Taijiquan organizations within the buffer zones, where there are 1237 public green spaces. (Figure 6). The results show that there are 148 Taijiquan organizations in the public green space, accounting for 71.15% of the total, which indicates that the Taijiquan organizations in London tend to develop in the surrounding area of public green space in London. There is obvious directivity of the distribution for Taijiquan organizations close to parks, green space, squares, and sports public venues in London.

## 5. Influencing Factors Analysis and Research Hypotheses

With factor analysis about the spatial distribution of Taijiquan organizations in London, research hypotheses are made to further explore the impact mechanism of the spatial distribution.

### 5.1. Analysis of Demographic Factors

#### 5.1.1. Influence of Population Density on the Spatial Distribution of Taijiquan Organizations

The population density reflects the degree of aggregation of the population. Generally speaking, the greater the population density is, the more service can be created, and the higher density of service industries can be [62]. Taijiquan organizations can be regarded as a kind of service industry in a sense. The people participating in the Taijiquan activities can be considered as consumers of Taijiquan culture. The more populated the places are, the greater potential of Taijiquan culture consumption there will be. Through the overlap analysis of the Spatial Distribution Map for Taijiquan Organizations in London and the Population Density Map of London (Figure 7), the number of Taijiquan organizations in all boroughs of London is positively correlated with their population density. Accordingly, a hypothesis is proposed as follows:

**Hypothesis 1** **(H1).**
*The population density of the boroughs in London is positively related to the number of Taijiquan organizations in the city.*


#### 5.1.2. Influence of Population Density of People over 65 Years Old on the Spatial Distribution of Taijiquan Organizations

Studies have shown that Taijiquan can bring the elderly good health benefits, including improved balance, attention, quality of sleep, reducing heart rate and blood pressure, increasing vagus nerve activities, lowering cholesterol, relieving chronic disease symptoms such as fiber muscle pain, osteoarthritis, and rheumatoid arthritis [2]. It is also shown that Taijiquan organizations in many countries and regions are mainly dominated by middle-aged and aged people [63,64]. The study uses demographic information about the elderly (aged over 65 years) in London released by the Office for National Statistics to create a vector map by means of ArcGIS 10.3 (Figure 8). As shown in the figure, the elderly (aged 65 years old or above) in London concentrate geographically in the dense core areas of the city rather than in the peripheral areas, which presents the positive correlation of the spatial distribution of Taijiquan organizations in London. Accordingly, a hypothesis is proposed as follows:

**Hypothesis 2** **(H2).***The population density of people aged 65 years old**or above in London is positively correlated to the number of Taijiquan organizations*.

#### 5.1.3. Influence of the Per Capita GDP on the Spatial Distribution of Taijiquan Organizations

Per capita GDP is an important indicator of the income and consumption capacity of residents, reflecting the social and economic status of the population in an area. Taijiquan organizations in London are mostly charitable organizations working for personal and public health benefits through the practice and participation of Taijiquan. Under the guidance of charity purposes and law in the UK, Taijiquan organizations offer both free and fee-paid Taijiquan courses according to the needs of customers at different levels and manage their finances for sustainable development. Therefore, it could be better for a Taijiquan organization to choose its location in an economically developed area. The study uses ArcGIS 10.3 to create a vector map according to the per capita GDP level in the boroughs of London (Figure 9). As shown in the figure, the per capita GDP level of the boroughs in the central part of London is much higher than that of the boroughs in the peripheral area of London, the positive correlation of the spatial distribution of Taijiquan organizations in London. Accordingly, a hypothesis is proposed as follows:

**Hypothesis 3** **(H3).***The higher the per capita income level of the borough, the more Taijiquan organizations there are*.

#### 5.1.4. Influence of the Average Education Level in Different Boroughs on the Spatial Distribution of Taijiquan Organizations

The average education level is an indicator of a civilized society and an important factor affecting how people accept and engage exotic cultures. A person’s education is also greatly related to their occupation, income, and social status. To a certain extent, it can affect one’s consumer behavior, the shaping of one’s outlook on life and values, and choices of lifestyle and new things. As Taijiquan has emigrated from China to Great Britain, a hypothesis is proposed as follows:

**Hypothesis 4** **(H4).***The average education level of different boroughs in London has a positive relationship with the number of Taijiquan organizations*.

### 5.2. Analysis of Public Facilities Accessibility

#### 5.2.1. Influence of Urban Traffic Accessibility on the Spatial Distribution of Taijiquan Organizations

Transportation infrastructure is fundamental for the development of an area [65]. An integrated traffic network consists of various transportation means to facilitate the flows of people, goods, information, and capital in the area. The location can also affect the traffic system. For example, one can find much easier access to transportation in the central area of a city than in the peripheral area. As the traffic network in inner London is more advanced than that in outer London, Taijiquan organizations with a good flow of people for public health benefits concentrate geographically in the core areas of inner London. Accordingly, a hypothesis is proposed as follows:

**Hypothesis 5** **(H5).**
*The more advanced the traffic system in an area is,*
*the more Taijiquan organizations there are.*


#### 5.2.2. Influence of Public Green Space Accessibility on the Spatial Distribution of Taijiquan Organizations

Several reviews show that green space accessibility and green park accessibility are significantly correlated with various health outcomes of urban residents [66,67,68]. As a fitness and health exercise, Taijiquan practice usually requires open space. Thus, Taijiquan organizations prefer public open space to private floor space for the sake of cost-saving. Accordingly, a hypothesis is proposed as follows:

**Hypothesis 6** **(H6).***The more public green**space in the borough, the more Taijiquan organizations there are*.

### 5.3. Influence of the Interaction between Demographic Factors and Public Facilities Accessibility on the Spatial Distribution of Taijiquan Organizations

The study aims to verify the interaction between the main effects of public facility accessibility and regulatory variables. Accordingly, hypotheses are proposed as follows:

**Hypothesis 7** **(H7).**
*The public green space accessibility is a more reliable indicator for the prediction of the number of Taijiquan organizations in different boroughs than urban traffic accessibility.*


**Hypothesis 8** **(H8).***There is an interaction between public green space accessibility and urban traffic accessibility, which means that there are more Taijiquan organizations in the boroughs with higher public green space accessibility and urban traffic accessibility*.

### 5.4. Verification Methods

Two prerequisites should be met for the verification of the spatial distribution of Taijiquan organizations in different boroughs in London:

(1) Maps for three major variables, namely, public facilities accessibility, the number of Taijiquan organizations in different boroughs, and regulatory variables (such as the average education level, per capita GDP, population density, etc.) must be available;

(2) The interaction between public facilities accessibility and regulatory variables should be clarified for the influence analysis of the variables (such as the number of Taijiquan organizations in boroughs). The study uses related verification methods to make responses to the hypotheses made above.

### 5.5. Variable Interpretations

#### 5.5.1. Dependent Variables

The number of Taijiquan organizations. At first, go to the official website of TCQUGB [46], choose the area of London, collect the zip codes of the Taijiquan organizations in the city, and finally calculate the number of Taijiquan organizations in different boroughs in London.

#### 5.5.2. Controlled Variables

Demographic variables can be used to predict the spatial distribution of Taijiquan organizations in different boroughs of London. Therefore, the study regards the average education level, per capita GDP level, population density, and population density of people over 65 years old as controlled variables.

#### 5.5.3. Independent Variables

The social and economic status of Taijiquan practitioners. Information about educational level, income level, and profession can be used as the measurement of socioeconomic status [69]. The study adopts the educational level as an independent variable.

Urban traffic accessibility. The study sets up a buffer zone of 200 m on both sides of the urban main traffic arteries for buffer analysis with ArcGIS10.3 and calculates the number of Taijiquan organizations within the buffer zone.

Public green space accessibility. The study uses the buffer analysis tool of the ArcGIS 10.3 with a [0-9]{1,}-m distance as a buffer radius to identify the number of Taijiquan organizations within the buffer zones and calculates the number of Taijiquan organizations within the buffer zone.

## 6. Results

The results are made by using population density, population density of the people above 65 years old, the average education level, per capita GDP level as controlled variables, and urban traffic accessibility and public green space accessibility as independent variables, and by applying Average education level × Urban traffic accessibility and Average education level × Public green space accessibility and Average education level × Urban traffic accessibility × Public green space as interactive variables in the multiple regression model. The results are shown in Table 2.

As shown in Table 2, demographic variables in Block 1 are explanatory for dependent variables (the number of Taijiquan organizations), as R^2^ = 0.438, F (4, 28) = 5.454, *p* 0.01. By applying the four controlled variables in the model, the total variation of Taijiquan organizations in London is 46.7%, β for the population density of people over 65 years old is 0.880 [T (28) = 2.56, *p* 0.05], indicating that the elderly (aged 65 years old or above) in London concentrate geographically in the dense core areas of the city rather than the peripheral area, which presents the positive correlation of the spatial distribution of Taijiquan organizations in London. Pearson correlation index is 0.599, which verifies Hypothesis 2. However, the population density of different boroughs in London does not significantly affect the number of Taijiquan organizations, the per capita GDP level and average education level does not significantly affect the number of Taijiquan organizations and there is no positive correlation between the per capita GDP level, average education level and the number of Taijiquan organizations. Thereby, Hypothesis 1, Hypothesis 3, and Hypothesis 4 are negated. 

By applying public facilities accessibility in Block 2 into the multiple regression model, the dependent variables can be expressed as R^2^ = 0.936, F (2, 26) = 63.762, *p* 0.001. Explanatory increment ΔR^2^ = 0.498, ΔF (2, 28) = 101.823, Δ*p* 0.001, showing the increasing expression and statistic significance of human geographic factors. That is, within the influence of demographic variables, the expression of public facilities accessibility can reach 47.1%. Among the four independent variables, the variable with the highest predictive ability is public green space accessibility (β = 0.587, *p* 0.001), followed by urban traffic (β = 0.520, *p* 0.05). It can be seen from the results: 1. The higher the urban traffic accessibility is, the more Taijiquan organizations there are, and there is a highly positive correlation between urban traffic accessibility and the number of Taijiquan organizations (Pearson correlation index is 0.932), which verifies H5; 2. The higher the public green space accessibility is, the more Taijiquan organizations there are, and there is a highly positive correlation between public green space accessibility and the number of Taijiquan organizations (Pearson correlation index is 0.941), which verifies H6; 3. With a comparison of public green space accessibility and urban traffic accessibility, and the Pearson correlation index for Taijiquan organizations, the public green space accessibility is a more reliable indicator for the prediction of the number of Taijiquan organizations in different boroughs than urban traffic accessibility, which verifies H7.

It is worth noting that the interpretation of three demographic variables in Block 1 is on the decline, as the value of population density drops from 0.662 to −0.251 [T (28) = −1.21, *p* = 0.237]; the value of population density of people over 65 years old drops from 0.880 [T (28) = −0.45, *p* = 0.659], the value of per capita GDP level drops from 0.844 to −0.179 [T (28) = −0.95, *p* = 0.351]. Although the average education level rises from −1.380 to 0.080 [T (28) = 0.69, *p* = 0.496], the four demographic variables are not sufficient to express the dependent variable (the number of Taijiquan organizations), and can only be explained as controlled variables in the model. It can be said that the interpretation of public facilities accessibility for calculating the number of Taijiquan organizations is made under the influence of demographic variables.

By applying interactive variables in Block 3 into the multiple regression model, the dependent variable can be expressed as ΔR^2^ = 0.010, ΔF (3, 23) = 1.413, Δ*p* = 0.265, with no statistical significance, as there is no interaction between public facilities accessibility and the number of Taijiquan organizations. Therefore, H8 is negated, which means that there is no interaction between education level and public facilities accessibility, and no interaction between public green space accessibility and urban traffic accessibility, and the number of Taijiquan organizations.

## 7. Discussion

This study conducts human geographical research on the spatial distribution of Taijiquan organizations in London, analyzing the influencing factors for the spatial distribution of Taijiquan organizations, and makes tentative steps for the global studies of Taijiquan.

Some studies think that the spatial layout of service industries is closely related to the directivity of urban traffic [70,71]. However, this study shows that Taijiquan organizations in London are not obviously distributed for access to the urban main traffic arteries, where there are more expensive venues for Taijiquan organizations that charge low fees for Taijiquan activities, and prefer quiet and open environments with fresh air. In other words, Taijiquan organizations are located for the consideration of costs and benefits, and traffic convenience is less considered. Previous studies also point out that public green space accessibility is significantly correlated with the health of the elderly [66,67,68]. The portion of green public space to the residential area within 1 km as a radius is correlated with the health of the people there [72]. This study finds that there is a correlation between public green space accessibility and the location of Taijiquan organizations. First, there is obvious directivity of the distribution for Taijiquan organizations close to parks, green space, squares, and sports public venues in London. Second, there is obvious directivity of the distribution for Taijiquan organizations close to the areas where the elderly concentrate geographically, just as Yang Chengfu (one of the forerunners of modern Yang style Taijiquan) stated, the practice of Taijiquan requires a quiet environment with fresh air that is open and ventilated [73].

According to the quantitative and qualitative research results, it can be inferred that there may be consistency in the influencing mechanism for the spatial distribution of Taijiquan organizations in developed cities. Comparative studies about cities at the same level can be launched for further research into the spatial distribution and influencing mechanisms for Taijiquan organizations.

Although the spatial distribution of Taijiquan organizations in London is not affected by the per capita GDP level and average education level of the people, can it be possible that Taijiquan organizations in cities of developing countries will be affected by these two factors in the same way? Comparative studies about cities at different levels can be launched to further research the spatial distribution and influencing mechanisms for Taijiquan organizations.

For people in London, Taijiquan culture is exotic, but how will it be transmitted into different cultural circles of the city? The follow-up research will pay attention to the spatial distribution and influencing mechanisms of Taijiquan in different cultural circles of the city.

## 8. Conclusions

The ratio of 208 Taijiquan organizations in inner London and outer London is 127: 81, the nearest neighboring index R is 0.68, the geographic concentration index G is 22.05 (the average distribution G is 6.3). The spatial distribution of Taijiquan organizations in London is of an agglomerated type. The distribution agglomeration of Taijiquan organizations in London is especially high in Camden, Islington, Barnett, and Wandsworth.

The rotation angle of the standard deviation ellipse is 112° with the elliptical long axis in the direction of northwest-southeast at a relatively small flattening rate, and the spatial distribution of Taijiquan organizations in London tending to expand in the direction of east and west.

There are only 117 Taijiquan organizations located near or along the urban main traffic arteries in London among all the 208 Taijiquan organizations, accounting for 56.25% of the total. It indicates that Taijiquan organizations in London are not obviously distributed for access to the urban main traffic arteries, which is also related to the feature of Taijiquan practitioners and the city they live in.

There are 148 Taijiquan organizations teaching in public green spaces, accounting for 71.15% of the total 208, which indicates that there is obvious directivity of the distribution for Taijiquan organizations close to parks, green space, squares, and sports public venues in London.

Among the four demographic factors (population density, population density of the people above 65 years old, the average education level, per capita GDP level), only the population density of people over 65 years old is positively related to the number of Taijiquan organizations in boroughs.

Urban traffic accessibility and public green space accessibility have a positive correlation with the spatial distribution of Taijiquan organizations, indicating that public facilities accessibility is an important factor affecting the distribution of Taijiquan.

In the relationship between public facilities accessibility, demographic factors, and the number of Taijiquan organizations and public facilities, there is no interaction between demographic factors (such as the average education level, per capita GDP, population density, etc.) and human geographic factors (urban traffic reaches and public green space accessibility).

## 9. Limitation

By means of the spatial analysis tools in ArcGIS 10.3(Environmental Systems Research Institute, Los Angeles, CA, USA) and SPSS 23.0(SPSS Inc., Chicago, IL, USA), this article maps out the spatial distributional pattern of the Taijiquan organizations in London and explores factors attributing to the spatial distribution of Taijiquan culture. The authors believe that all these opinions may be related to London where they live, which is one of the most developed cities in the world with a high per capita GDP level and education level according to the *Kearney 2019 Global Cities Report*. However, due to the limited choice of Taijiquan organizations and demographic factors in London, the study did not investigate the differences between Taijiquan organizations. It is also shown in the study that there is no correlation between the per capita GDP level and the average education level and the number of the Taijiquan organizations in London, which is contrary to other research results that people at a higher education level have stronger purchase capacity and intentions. In follow-up research, it is necessary to expand the research to other global cities or similar organizations. As such, this article is the first output of an emerging study mapping the distribution of Taijiquan organizations in a global city.

## Figures and Tables

**Figure 1 ijerph-18-08452-f001:**
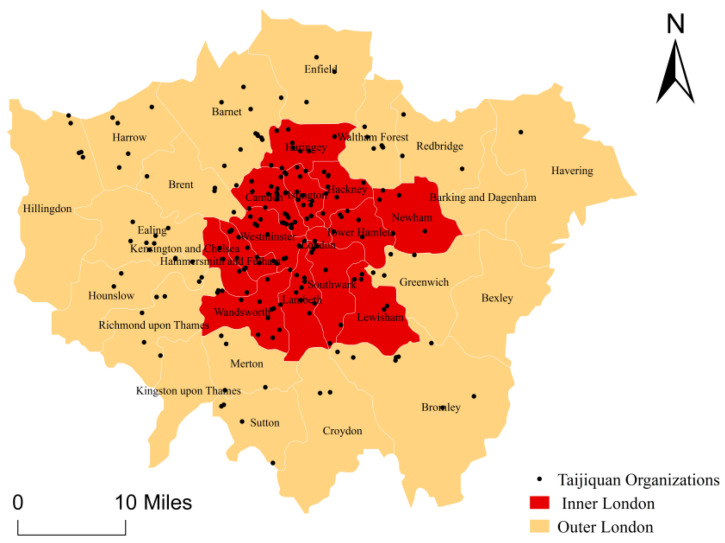
Distribution Map of Taijiquan Organizations in Inner and Outer London.

**Figure 2 ijerph-18-08452-f002:**
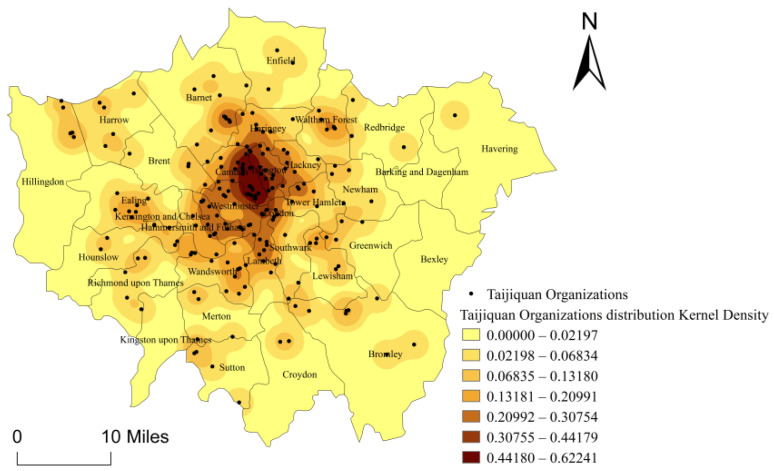
Kernel Density Map for the Distribution of Taijiquan Organizations in London.

**Figure 3 ijerph-18-08452-f003:**
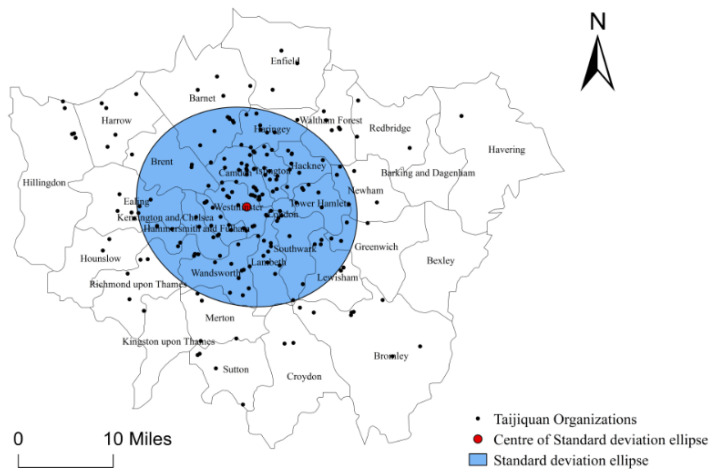
Direction Distribution Map for Taijiquan Organizations in London.

**Figure 4 ijerph-18-08452-f004:**
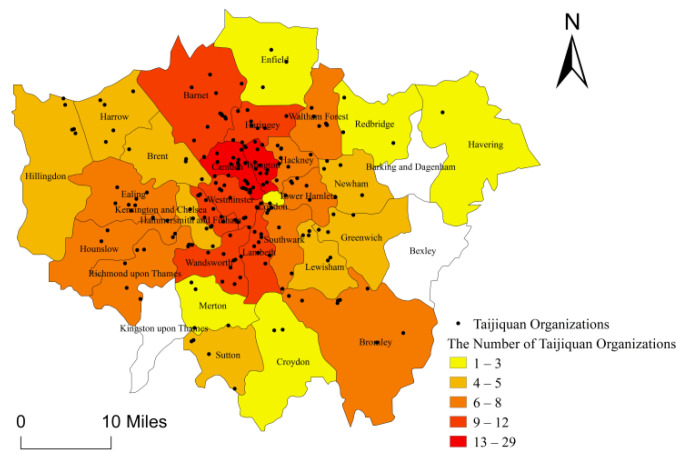
Distribution Density Map of Taijiquan Organizations in Boroughs of London.

**Figure 5 ijerph-18-08452-f005:**
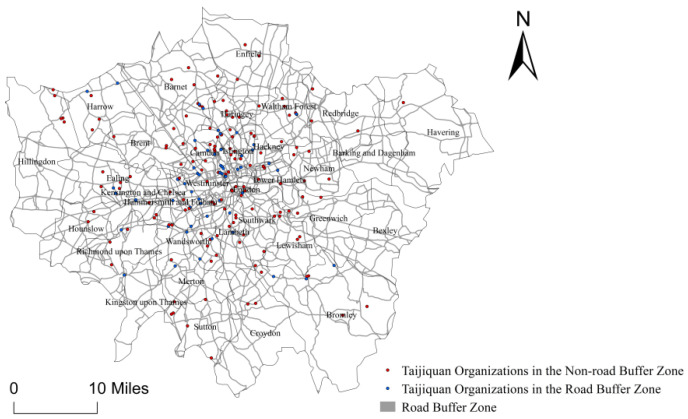
Main Traffic Route Map for the Spatial Distribution of Taijiquan Organizations in London.

**Figure 6 ijerph-18-08452-f006:**
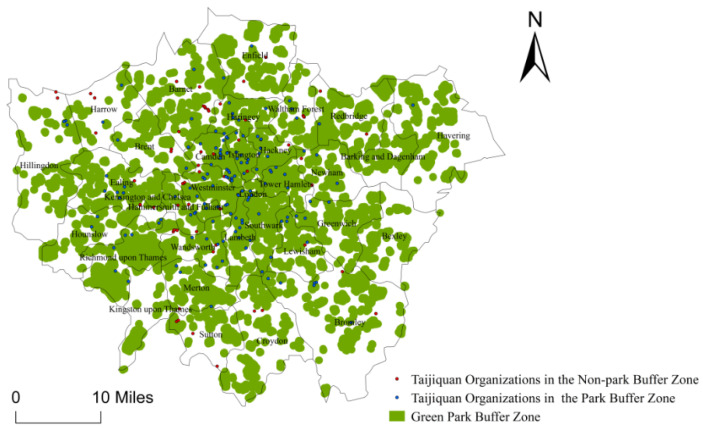
Green Space Map for the Spatial Distribution of Taijiquan Organizations in London.

**Figure 7 ijerph-18-08452-f007:**
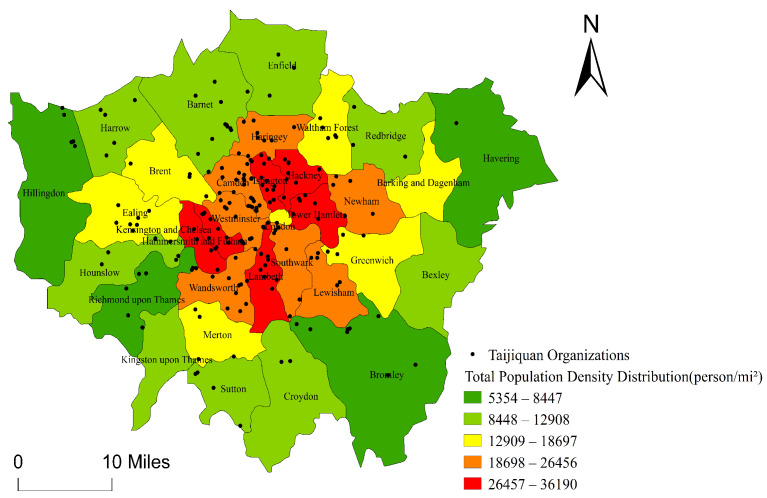
Population Density Map of Boroughs in London.

**Figure 8 ijerph-18-08452-f008:**
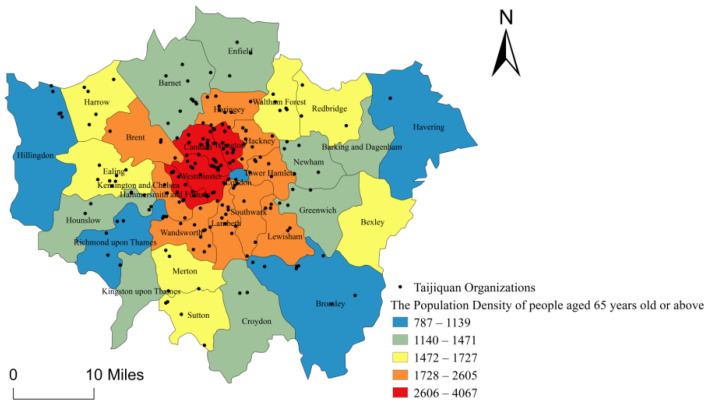
Population Density Map of People Aged 65 Years Old or Above in London.

**Figure 9 ijerph-18-08452-f009:**
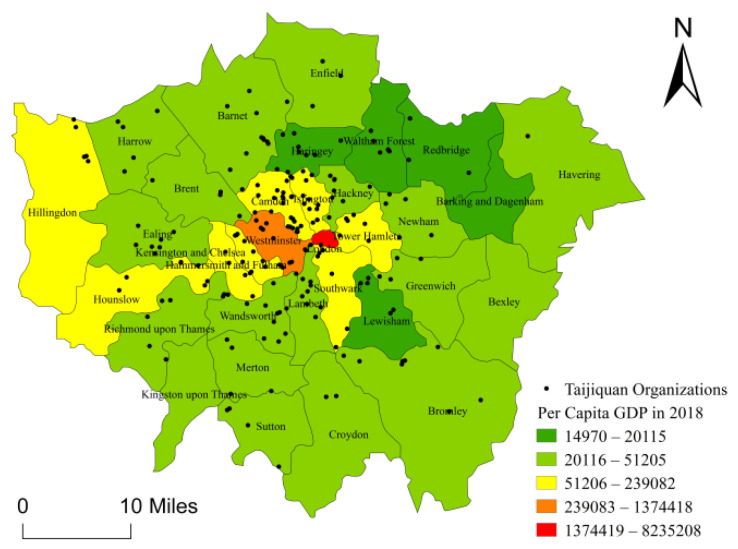
Per Capita GDP Distribution Map of Boroughs in London.

**Table 1 ijerph-18-08452-t001:** Statistics on the Number of Taijiquan Organizations in the Boroughs of London.

Borough	Number	Rank	Percentage	Accumulated Percentage
Camden	25	1	12.02%	12.02%
Islington	15	2	7.21%	19.23%
Barnet	12	3	5.77%	25.00%
Wandsworth	12	3	5.77%	30.77%
Kensington and Chelsea	11	5	5.29%	36.06%
City of Westminster	10	6	4.81%	40.87%
Haringey	9	7	4.33%	45.19%
Lambeth	9	7	4.33%	49.52%
Bromley	8	9	3.85%	53.37%
Ealing	7	10	3.37%	56.73%
Hackney	7	10	3.37%	60.10%
Hounslow	7	10	3.37%	63.46%
Tower Hamlets	7	10	3.37%	66.83%
Richmond upon Thames	6	14	2.88%	69.71%
Southwark	6	14	2.88%	72.60%
Waltham Forest	6	14	2.88%	75.48%
Hammersmith and Fulham	5	17	2.40%	77.88%
Harrow	5	17	2.40%	80.29%
Hillingdon	5	17	2.40%	82.69%
Lewisham	5	17	2.40%	85.10%
Newham	5	17	2.40%	87.50%
Sutton	5	17	2.40%	89.90%
Brent	4	23	1.92%	91.83%
Greenwich	4	23	1.92%	93.75%
Enfield	3	25	1.44%	95.19%
Merton	3	25	1.44%	96.63%
Redbridge	3	25	1.44%	98.08%
Croydon	2	28	0.96%	99.04%
Havering	1	29	0.48%	99.52%
City of London	1	29	0.48%	100.00%
Bexley	0	31	0.00%	100.00%
Barking and Dagenham	0	31	0.00%	100.00%
Kingston upon Thames	0	31	0.00%	100.00%

**Table 2 ijerph-18-08452-t002:** Multivariate Hierarchical Regression Analysis of Public Facilities Accessibility for the Spatial Distribution of Taijiquan Organizations.

Variables in the Model		Model 1			Model 2			Model 3		
		Beta	t	Sig.	Beta	t	Sig.	Beta	t	Sig.
Controlled Variables	Block 1									
	Population density	0.662	1.77	0.087	−0.251	−1.21	0.237	−0.200	−0.96	0.350
	Population density of people over 65 years old	0.880 **	2.56	0.016	−0.070	−0.45	0.659	0.124	0.67	0.508
	Average education level	−1.380	−1.92	0.080	0.193	0.69	0.496	0.041	0.13	0.901
	Per capita GDP level	0.844	1.72	0.096	−0.179	−0.95	0.351	−0.058	−0.28	0.780
Independent variables	Block 2									
	Urban traffic accessibility				0.570 ***	3.66	0.001	1.208 *	2.4	0.043
	Public green space accessibility				0.517 ***	4.66	0.000	0.377	0.60	0.556
Interactive variables	Block 3									
	Education level × urban traffic accessibility							−0.761	−1.16	0.257
	Education level × Public green space accessibility							0.313	0.40	0.693
	Educational level × Urban traffic accessibility × public green space accessibility							−0.160	−0.90	0.377
Model Abstract	R^2^	0.438			0.936			0.946		
	F	5.454 **			63.762 ***			45.003 ***		
	*p*	0.002			0.000			0.000		
	ΔR^2^	0.438			0.498			0.010		
	ΔF	5.454 **			101.823 ***			1.413		
	Δ*p*	0.002			0.000			0.265		

Note: Sig. is Significance; * *p* 0.05, ** *p* 0.01, *** *p* 0.001, and β is a standard coefficient.

## Data Availability

The data presented in this study are available on request from the corresponding author, which is not accessible to the wider public.

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
