# Peer review of "From Green Spaces to Squares: Mapping the Distribution of Taijiquan Organizations in London"

_ijerph, 2021, doi:10.3390/ijerph18168452_

Round 1
Reviewer 1 Report
Dear Authors,
I read with interest the article that describes the important research problem of Taijiquan. The article is based on reliable empirical research. The value of work is a valuable set of maps illustrating the determinants and background of the development of the Taijiquan network.
However, I have two remarks on the article.
The first is the need to expand the chapter on research review. Of particular importance would be the development of Taijiquan in other cities outside of China or other similar organizations. In addition, the issue of literature on green areas in large cities and their role in recreation is also important.
The second problem I see in the article is a very narrow discussion. The authors carried out ambitious and interesting research. It would be worth discussing this topic in broader contexts (other publications on this topic or similar) and terms of your achievements (weaknesses, strengths, challenges). Therefore, I suggest separating the discussion from the results and a more in-depth reflection on Taijiquan in the aforementioned contexts.
Author Response
Dear Reviewer 1,
Thank you for your kind comments on our article, which were very precise and helpful for us. In accordance with your suggestions, we have made a substantial revision of this article. Below, you will find a breakdown of the responses to your comments (in italics):
The first is the need to expand the chapter on research review. Of particular importance would be the development of Taijiquan in other cities outside of China or other similar organizations. In addition, the issue of literature on green areas in large cities and their role in recreation is also important.
Thank you so much for all your valuable advices, we have made further modifications in the text which are marked in red. We have now added various citations within the paragraphs that you have identified, especially the development of Taijiquan in other cities outside of China and the literature on green areas and their role in recreation you mentioned in your review.
The second problem I see in the article is a very narrow discussion. The authors carried out ambitious and interesting research. It would be worth discussing this topic in broader contexts (other publications on this topic or similar) and terms of your achievements (weaknesses, strengths, challenges). Therefore, I suggest separating the discussion from the results and a more in-depth reflection on Taijiquan in the aforementioned contexts.
We appreciate your suggestion, and we have made correction according to your comments and suggestions. Section 7 and Section 8 are revised to explain the topic in broader contexts. Section 9 separately expresses a reflection about the limitation and challenges.
We tried our best to improve the manuscript and made some changes in the manuscript. These changes will not influence the content and framework of the paper. And here we did not list the changes but marked them in red in the revised version of the paper.
We hope you are pleased with the revised manuscript, which also includes the changes expected from other peer reviewers.
Thanks again, and kind regards.
Reviewer 2 Report
L37 "the popularization and the globalization of Taijiquan are in urgent need": can you justify this statement? In which sense they are "needed"?
L47 "the real demand for China's increasing cultural influence and competitiveness.": demand from whom?
L64 "The global study on Taijiquan [...] analyzes the spatial distribution of Taijiquan organizations in London": is the London case significant as a template for the global distribution of Taijiquan?
L75 "Since the 18th century, London has been the world's most important political, economic, cultural, art, entertainment and sports event center": I would say "one of the most ... centers": other cites have also been important.
L96 "Especially those in London have demonstrated representative spatial distribution patterns and features.": is the existence of these patterns known in literature or it has emerged during this research (in the latter case it is a circular reference: London has been selected for the presence of pattern, which have been detected by this study after the selection),
L104 "We use data and resources from the official website of Tai Chi & Qigong Union for Great Britain(TCUGB) [31] and to identify the number and coordinates" -> "We use data and resources from the official website of Tai Chi & Qigong Union for Great Britain(TCUGB) [31] to identify the number and coordinates" (removed "and" after [31]").
L110 "and add the date": the data?
L116 "we converted the coordinates in the network map into geographic coordinates": which "network" map? Why do coordinates need to be converted? Are they expressed with respect to different datum/projections?
L123 Section 3.1: it would be useful to specify that some analyses provide global indexes and some create maps.
L127 "(BUFFER)": is this the name of the procedure in ArcGIS?
L133 "Nearest Neighbor Index (NNA)": NNI? ArcGIS manual uses "Average Nearest Neighbor (ANN)"
L134 "[Nearest Neighbor Index] is an indicator that measures two punctiform elements adjacent": in which sense it measures two elements?
From the ArcGIS manual (online at https://desktop.arcgis.com/en/arcmap/10.3/tools/spatial-statistics-toolbox/average-nearest-neighbor.htm):
"The Nearest Neighbor Index is expressed as the ratio of the Observed Mean Distance to the Expected Mean Distance. The expected distance is the average distance between neighbors in a hypothetical random distribution."
L135 What is a "geospatial space"?
L136 What are "punctiform expressions"? Point distributions?
L137 Eq. 1. It must be specified that the bar over r_1 and r_E indicates the mean values for all the points in the map.
The third and fourth parts of the equation are wrong, r_1 or an equivalent quantity is missing, see https://pro.arcgis.com/en/pro-app/latest/tool-reference/spatial-statistics/h-how-average-nearest-neighbor-distance-spatial-st.htm and your [36], page 445.
L139 "r_E is the most adjacent distance in theory" -> "r_E is the expected distance for a random distribution" (or r_E with the bar is the expected mean distance for a random distribution).
"m is the number of points in the formula": in the map.
L140 "R For the nearest index": R is the Nearest Neighbor Index.
L150 "in i borough of London" -> "in the i-th borough of London"
L161 "distance from the estimation point x to the event X_i": what is the "event X_i"?
L168 "The study applies buffer analysis to the calculation about public traffic arteries and public green space in London.": the calculation of what?
L172 "Direction distribution is used to calculate the standard distance in the directions of X and Y with the average center of the set of points as the starting point.": distance of what?
L176 Is an English version of [41] (Wang Juehan, Zhou Chunshan. Spatial Distribution and Its Influential Factors of Commercial Fitness Clubs in Guangzhou. Tropical Geography, 2018, 38 (1): 120-130.DOI: 10.13284/j.cnki.rddl.002979) available? Is it possible to cite a paper in English instead?
L186 "and may be subject to variables.": explanatory variables?
L225 "The average distance of ideal spatial distribution": in which sense "ideal"? Corresponding the evenly distributed points?
L227 "which shows the spatial distribution of Taijiquan organizations in London is" -> "which shows that the spatial distribution of Taijiquan organizations in London is" (added "that").
L232 "is high with Camden" -> "is high in Camden"?
L244 Section 4.4: is a true network analysis tool available, to avoid the use of a distance reduction coefficient? Which value of the range 1.2 - 1.4 has been used (900/700=1.29 but I guess 1.3 has been used and the distance has been rounded up to 700m)?
Footnote of page 8: what are "regional regions"?
L293 "concentrate geographically in the dense core areas of the city other than the peripheral area" -> "concentrate geographically in the dense core areas of the city rather than in the peripheral areas".
L326 "As Taijiquan has immigrated from China to Great Britain": emigrated?
L335 "The location quotient can also affect traffic system": what is the "location quotient"?
L365 "Three major variables, namely, public facilities accessibility, the number of Taijiquan organizations in different boroughs and regulatory variables (such as the average education level, per capita GDP, population density, etc.);": the verb is missing in this phrase. Maybe it can be "Maps for three major variables, namely [...] must be available;"
L396-401 This is a repetition of sections 5.5.2 and 5.5.3.
L403 Table 2: what are models 1,2 and 3? What is "Sig.", significance? Describe in the caption.
The caption has a spurious "." in " for.the".
L409 "βfor" -> "β for"
L411 "other" -> rather
L413 The parentheses should be deleted.
L445 I do not understand the phrase "It can be said that the interpretation of public facilities accessibility for calculating the number of Taijiquan organizations obtained under the influence of demographic variables.": is part of the sentence missing?
L462 "Previous studies also point out that public green space accessibility is significantly correlated with the health of the elderly [48-50]. It is also verified in the study": not really. You found a correlation between public green space accessibility and the location of Taijiquan organizations. There may be a (very) indirect link.
L472 "London [...] which is a most developed city in the world": citation needed.
Author Response
Dear Reviewer 2,
Thank you for your kind comments on our article, which were very precise and helpful for us. In accordance with your suggestions, we have made a substantial revision of this article. Below, you will find a breakdown of the responses to your comments (in italics):
L37 "the popularization and the globalization of Taijiquan are in urgent need": can you justify this statement? In which sense they are "needed"?
We appreciate your suggestion, and we have made corrections in the text. Delete “urgent”. A statement from Ryan is added as the study is made to call more attention to the development of Taijiquan in other cities outside of China or other similar organizations. Some literature review is added in Introduction and Section 2.1
L47 "the real demand for China's increasing cultural influence and competitiveness.": demand from whom?
Thanks. Confirmed and deleted the part. It is not the point to be explained in the paper.
L64 "The global study on Taijiquan [...] analyzes the spatial distribution of Taijiquan organizations in London": is the London case significant as a template for the global distribution of Taijiquan?
Thanks. We have made corrections according to your comments and suggestions. London case is significant as a template for the global distribution of Taijiquan. Reasons are added in 2.1. Research area selection.
L75 "Since the 18th century, London has been the world's most important political, economic, cultural, art, entertainment and sports event center": I would say "one of the most ... centers": other cites have also been important.
Thanks. We have made corrections according to your comments and suggestions. Confirmed and changed into “one of the most... centers”. Please see L95.
L96 "Especially those in London have demonstrated representative spatial distribution patterns and features.": is the existence of these patterns known in literature or it has emerged during this research (in the latter case it is a circular reference: London has been selected for the presence of pattern, which have been detected by this study after the selection),
According to your comment and suggestion. We have deleted “Especially those in London have demonstrated representative spatial distribution patterns and features”.
L104 "We use data and resources from the official website of Tai Chi & Qigong Union for Great Britain(TCUGB) [31] and to identify the number and coordinates" -> "We use data and resources from the official website of Tai Chi & Qigong Union for Great Britain(TCUGB) [31] to identify the number and coordinates" (removed "and" after [31]").
We appreciate your suggestion, and we have made corrections in the text. Please see L127.
L110 "and add the date": the data?
We appreciate your suggestion, and we have made corrections in the text. Please see L132.
L116 "we converted the coordinates in the network map into geographic coordinates": which "network" map? Why do coordinates need to be converted? Are they expressed with respect to different datum/projections?
Thanks. Confirmed and changed into “converted the valid zip codes of Taijiquan organizations into geographic coordinates”. Please see L139
L123 Section 3.1: it would be useful to specify that some analyses provide global indexes and some create maps.
Thank you for your kindly reminder. Some references about ArcGIS 10.3 and SPSS 23.0 are added in Section 3.1 and Section 3.2
L127 "(BUFFER)": is this the name of the procedure in ArcGIS?
Thanks. Delete “(BUFFER)”. Please see L149.
L133 "Nearest Neighbor Index (NNA)": NNI? ArcGIS manual uses "Average Nearest Neighbor (ANN)"
We are very sorry for our negligence, and we have fixed it. Please see L156.
L134 "[Nearest Neighbor Index] is an indicator that measures two punctiform elements adjacent": in which sense it measures two elements?
From the ArcGIS manual (online at https://desktop.arcgis.com/en/arcmap/10.3/tools/spatial-statistics-toolbox/average-nearest-neighbor.htm):
"The Nearest Neighbor Index is expressed as the ratio of the Observed Mean Distance to the Expected Mean Distance. The expected distance is the average distance between neighbors in a hypothetical random distribution."
We have made correction according to your comments and suggestions. Please see L156-L159.
L135 What is a "geospatial space"?
Thanks. Changed into “geological space”. Please see L160.
L136 What are "punctiform expressions"? Point distributions?
Thanks. Confirmed and changed. Please see L160.
L137 Eq. 1. It must be specified that the bar over r_1 and r_E indicates the mean values for all the points in the map. The third and fourth parts of the equation are wrong, r_1 or an equivalent quantity is missing, see https://pro.arcgis.com/en/pro-app/latest/tool-reference/spatial-statistics/h-how-average-nearest-neighbor-distance-spatial-st.htm and your [36], page 445.
We have made correction according to your comments and suggestions. Please see L163-L168.
L139 "r_E is the most adjacent distance in theory" -> "r_E is the expected distance for a random distribution" (or r_E with the bar is the expected mean distance for a random distribution)."m is the number of points in the formula": in the map.
We have made correction according to your comments and suggestions. Please see L165.
L140 "R For the nearest index": R is the Nearest Neighbor Index.
We appreciate your suggestion, and we have made corrections in the text. Please see L166.
L150 "in i borough of London" -> "in the i-th borough of London"
We appreciate your suggestion, and we have made corrections in the text. Please see L178.
L161 "distance from the estimation point x to the event X_i": what is the "event X_i"?
We appreciate your suggestion, and we have made corrections in the text. We Changed into “ is the distance from the estimation point to the actual point . ”. Please see L188.
L168 "The study applies buffer analysis to the calculation about public traffic arteries and public green space in London.": the calculation of what?
We appreciate your suggestion, and we have made corrections in the text. We have Changed into “calculation the number of taijiquan organizations”. Please see L196.
L172 "Direction distribution is used to calculate the standard distance in the directions of X and Y with the average center of the set of points as the starting point.": distance of what?
We appreciate your suggestion, and we have made corrections in the text. We have Changed into “distance between two focal points. Please see L200.
L176 Is an English version of [41] (Wang Juehan, Zhou Chunshan. Spatial Distribution and Its Influential Factors of Commercial Fitness Clubs in Guangzhou. Tropical Geography, 2018, 38 (1): 120-130.DOI: 10.13284/j.cnki.rddl.002979) available? Is it possible to cite a paper in English instead?
Thanks. It is a Chinese paper. And it is also found in the Arc GIS manual (online at:
https://pro.arcgis.com/en/pro-app/2.6/tool-reference/spatial-statistics/h-how-directional-distribution-standard-deviationa.htm. Please see L205
L186 "and may be subject to variables.": explanatory variables?
Thanks. Confirmed and changed. Please see L216.
L225 "The average distance of ideal spatial distribution": in which sense "ideal"? Corresponding the evenly distributed points?
Thanks. Changed into “expected”. Please see L254.
L227 "which shows the spatial distribution of Taijiquan organizations in London is" -> "which shows that the spatial distribution of Taijiquan organizations in London is" (added "that").
Thank you so much, we have made a correction in the article. Please see L257.
L232 "is high with Camden" -> "is high in Camden"?
Thank you so much, we have made a correction in the article. Please see L262.
L244 Section 4.4: is a true network analysis tool available, to avoid the use of a distance reduction coefficient? Which value of the range 1.2 - 1.4 has been used (900/700=1.29 but I guess 1.3 has been used and the distance has been rounded up to 700m)?
Thank you so much, we have made a correction in the article. Changed into “average”. Please see L281.
Footnote of page 8: what are "regional regions"?
Thank you so much, we have made a correction in the article. Changed into “This study uses public green space to refer such areas as parks, green spaces, squares, sports public venues, ect. .”
L293 "concentrate geographically in the dense core areas of the city other than the peripheral area" -> "concentrate geographically in the dense core areas of the city rather than in the peripheral areas".
Thank you so much, we have made a correction in the article. Please see L325.
L326 "As Taijiquan has immigrated from China to Great Britain": emigrated?
Thank you so much, we have made a correction in the article. Please see L358.
L335 "The location quotient can also affect traffic system": what is the "location quotient"?
Thanks. Mistyping. Delete "quotient". Please see L367.
L365 "Three major variables, namely, public facilities accessibility, the number of Taijiquan organizations in different boroughs and regulatory variables (such as the average education level, per capita GDP, population density, etc.);": the verb is missing in this phrase. Maybe it can be "Maps for three major variables, namely [...] must be available;"
Thank you so much, we have made a correction in the article. Please see L397-L399.
L396-401 This is a repetition of sections 5.5.2 and 5.5.3.
Thanks. As interactive variables in the multiple regressions model are added in the analysis, the repetitive description is needed.
L403 Table 2: what are models 1,2 and 3? What is "Sig.", significance? Describe in the caption.
Thanks. Three models are listed according to the different considerations about different variables. Sig. is Significance. Please see L437.
The caption has a spurious "." in " for.the".
The caption has a spurious "." in " for.the". Please see L435.
L409 "βfor" -> "β for"
Thank you so much, we have made a correction in the article. Please see L441.
L411 "other" -> rather
Thank you so much, we have made a correction in the article. Please see L444.
L413 The parentheses should be deleted.
Thank you so much, we have made a correction in the article. Please see L445.
L445 I do not understand the phrase "It can be said that the interpretation of public facilities accessibility for calculating the number of Taijiquan organizations obtained under the influence of demographic variables.": is part of the sentence missing?
We appreciate your suggestion, and we have made corrections in the text. “obtained” is changed into “is made”. Please see L479.
L462 "Previous studies also point out that public green space accessibility is significantly correlated with the health of the elderly [48-50]. It is also verified in the study": not really. You found a correlation between public green space accessibility and the location of Taijiquan organizations. There may be a (very) indirect link.
Thank you so much, we have made a correction in the article.
L472 "London [...] which is a most developed city in the world": citation needed.
Thank you so much, we have made a correction in the article.
We tried our best to improve the manuscript and made some changes in the manuscript. And here we did not list the changes but marked them in red in the revised version of the paper. We really hope you are pleased with the revised manuscript, which also includes the changes expected from other peer reviewers.
Thanks again for your precise and comments, and kind regards.
Round 2
Reviewer 1 Report
Dear Authors,
Thank you to the Authors for improving the article. I have one more comment on the new version of the article. I believe that the new chapter: Limitation, should be placed before the Conclusions chapter.